# Evaluation of the Terrestrial $^{222}$Rn Flux from $^{210}$Pb Deposition Measurements

**Mauro Magnoni \*, Luca Bellina, Stefano Bertino, Brunella Bellotto and Enrico Chiaberto**

ARPA Piemonte, Department of Physical and Technological Risks, 10015 Ivrea, TO, Italy;
l.bellina@arpa.piemonte.it (L.B.); s.bertino@arpa.piemonte.it (S.B.); b.bellotto@arpa.piemonte.it (B.B.);
e.chiaberto@arpa.piemonte.it (E.C.)
**\*** Correspondence: m.magnoni@arpa.piemonte.it

**Abstract:** The study of the $^{222}$Rn terrestrial flux (Bq/(m$^2$·s) or Bq/(m$^2$·h)) is a complex issue involving both radiation-protection and environmental aspects. While the radiation-protection aspects are quite obvious—it has been well known for several decades that soil is the major source of indoor radon—environmental issues such as the correlation with conventional pollutants (PM$_{2.5}$, PM$_{10}$, NOX, etc.) and the use of radon for the esmation of the natural component of GHG (CO$_2$) emissions are relatively less discussed in spite of their growing relevance. In this work we present a method for the estimation of the average value of $^{222}$Rn flux from HPGe γ-spectrometry $^{210}$Pb measurements performed on wet and dry deposition samples gathered monthly in the period 2006–2020. The results obtained with this technique give an average radon flux in the period $\Phi = 57 \pm 27$ Bq/(m$^2$·h), the value of which is comparable with those coming from other methods and direct radon flux measurements as well. The method can thus be used to obtain a worldwide map of the radon flux.

**Keywords:** terrestrial radon flux; environmental radioactivity monitoring network; $^{210}$Pb deposition; HPGe γ-spectrometry





## 1. Introduction

The measurement of the terrestrial radon flux is an important and well known issue for radiation protection: many studies have demonstrated that the radon flux coming from the ground is by far the most important contribution to the radon levels found in dwellings and workplaces [1–5]. This fact has also been explicitly recognized by many legislations. At the European level, for example, a directive was issued, the 59/2013/Euratom [6], in which each EU member state has a mandate for the individuation and definition of *radon priority areas*, i.e., areas where the radon flux from the ground is significantly greater than the average. Besides this, many other scientific studies deal with radon flux: many researchers have investigated the correlation of radon flux variations with seismic and volcanological phenomena [7–12]. The knowledge of local radon flux values is very important for atmospheric studies as well, aiming to evaluate the motion and the origin of air masses [13–16]. Radon flux data and measurements are also used for the forecast and estimation of the occurrence of very high concentrations of some conventional pollutants, such as PM$_{10}$, PM$_{2.5}$, NOX, and benzene, during particular meteorologic conditions, often characterized by thermal inversions [17–23].

More recently, a growing interest in radon flux measurements has arisen among researchers trying to evaluate the natural component of greenhouse gases; indeed, the radon flux can be used as a proxy for the estimation of the terrestrial CO$_2$ flux [24–28]. In all these fields of study, the knowledge of the values of the terrestrial radon flux $\varphi$ is crucial. Unfortunately, this kind of measurement is far from simple to carry out and currently suffers from a great lack of standardization. Very different approaches have been proposed, using many different instruments and detectors. One of the main problems of a direct radon flux measurement is the difficulty in detecting radon coming from the ground without

perturbing the exhalation process in a substantial way. Recently, an ongoing European project (traceRadon, [29]) has been trying to tackle these problems by promoting large intercomparison programs of different devices able to follow the time evolution of the radon exhalation rate in different soils.

In this work a new and different method is proposed, based on [210]Pb measurements. The evidence of an excess [210]Pb flux of atmospheric origin has been investigated in several works dealing with the sediment accumulation rate and sediment core chronology studies; see, for example [30]. This [210]Pb flux, if properly measured, can also be used to estimate the average value of the terrestrial radon flux indirectly. The method proposed derives the radon flux from an historical series of [210]Pb γ spectrometry deposition measurements performed in the framework of the RESORAD network, the Italian National Radioactivity Monitoring Network.

## 2. Materials and Methods

After being produced by radioactive decay, radon is separated from its parent [226]Ra and, because of its relatively long half-life (3.82 days), is released into the atmosphere, giving rise to the lower part of the uranium natural radioactive series which substantially contributes to atmospheric radioactivity (see Figure 1). Let us consider in particular the system consisting of six radionuclides, circled in red in Figure 1, having [222]Rn as parent and ending to [210]Pb. [210]Pb is produced by the decay of [214]Po, the last of the so called *short-lived radon daughters*, the others being [218]Po, [214]Pb, [214]Bi, usually named short lived radon daughters because of their very short half lives. [210]Pb is a β emitter having a quite long half-life, 22.23 years, emitting a relatively soft β radiation (end-point energy 63.5 keV, [31]), followed almost immediately by a low energy γ line (46.5 keV) coming from the de-excitation of the 0-excited state of [210]Bi.

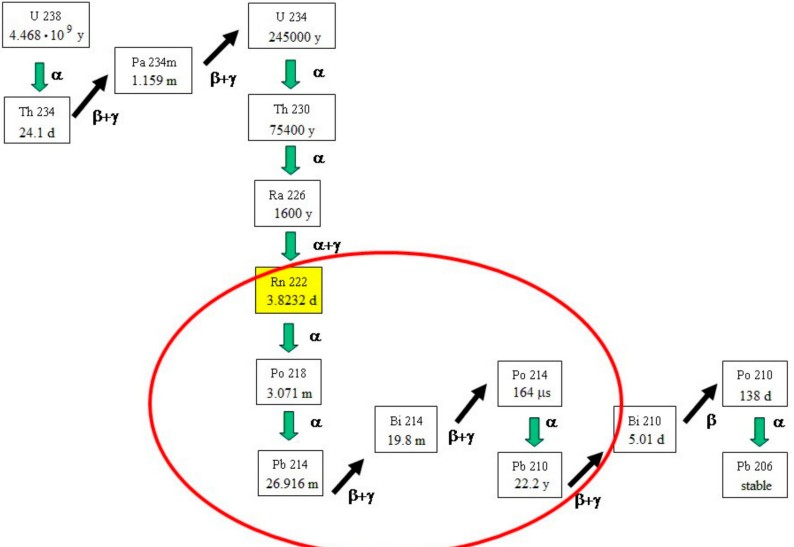

**Figure 1.** The lower part of the uranium series in atmosphere: the six radionuclides considered in this work are circled in red.

It can be observed that, for practical reasons, this system can be simplified and considered substantially equivalent to a five-radionuclide system. Indeed, due to the very short half-life of [214]Po (164 µs), which assures an almost immediate achievement of the secular equilibrium condition between [214]Bi and [214]Po, the two radionuclides evolve in time together and can thus be considered as a unique radionuclide emitting, almost at the same time, both α and β radiation. In the following, in order to distinguish the different radionuclides belonging to the system, they will be indicated with the following indexes:

$j = 0$ for $^{222}$Rn, $j = 1$ for $^{218}$Po, $j = 2$ for $^{214}$Pb, $j = 3$ for the coupled system $^{214}$Bi/$^{214}$Po, and $j = 4$ for $^{210}$Pb.

If we assume that the terrestrial radon flux is given by $\varphi$, expressed as an atomic flux (atoms/(m$^2$·s) or atoms/(m$^2$·h) while $D_j$ is the corresponding quantities of radionuclides that build up in the whole atmosphere (atoms/m$^2$), the following five equations may be written:

$$\frac{dD_0}{dt} + \lambda_0 D_0 = \phi$$

$$\frac{dD_{j+1}}{dt} + (\lambda_{j+1} + \lambda_d)D_{j+1} = \lambda_i D_i \qquad j = 0, 1, 2, 3$$

where $\lambda_j$ is the decay constant of the $^{222}$Rn and its progeny and $\lambda_d$ is the removal rate of the daughters from the atmosphere, due to wet and dry deposition processes, the value of which is assumed equal for each radionuclide.

The solutions of these equations are straightforward, while resulting in quite complicated expressions. However, these expressions can be dramatically simplified considering the corresponding asymptotic solutions, i.e., the solutions obtained for t→∞. The asymptotic solutions provide very simple and time independent expressions that are much easier to handle. Indeed, it can be easily demonstrated that the exact time dependent solutions differ from the asymptotic expressions by transient factors that became negligible very quickly—in a few hours, in accordance with the half-lives of the short lived radon daughters. This does not happen with the last equation, which referred to $^{210}$Pb: in this case the transient is a little longer, a few days, being of the order of $1/\lambda_d$. An estimation of the $\lambda_d$ value will be given in the next sections of this paper.

The asymptotic solutions of the system, expressed in term of activities (inventories), being $A_j = \lambda_j D_j$ (Bq/m$^2$), are the following:

$$A_0 = \phi \tag{1}$$

$$A_1 = \frac{\lambda_1 \phi}{\lambda_1 + \lambda_d}$$

$$A_2 = \frac{\lambda_1 \lambda_2 \phi}{(\lambda_1 + \lambda_d) \times (\lambda_2 + \lambda_d)}$$

$$A_3 = \frac{\lambda_1 \lambda_2 \lambda_3 \phi}{(\lambda_1 + \lambda_d) \times (\lambda_2 + \lambda_d) \times (\lambda_3 + \lambda_d)}$$

$$A_4 = \frac{\lambda_1 \lambda_2 \lambda_3 \lambda_4 \phi}{(\lambda_1 + \lambda_d) \times (\lambda_2 + \lambda_d) \times (\lambda_3 + \lambda_d) \times (\lambda_4 + \lambda_d)}$$

The last equation is the only relevant expression for our purposes, giving the $^{210}$Pb atmospheric inventory in the whole atmosphere column (Bq/m$^2$) as a function the radon flux $\varphi$. Thus, in order to estimate $\varphi$, the $^{210}$Pb atmospheric inventory $A_4$, must be related to a measurable quantity, i.e., to the wet and dry deposition, often indicated also as the fallout. The fallout measurements of airborne radionuclides are one of the most important pillars of any environmental radioactivity network, being one of the most sensitive measurement techniques, although relatively simple to perform. From decades, routine fallout measurements were performed monthly in our laboratory (Ivrea, northwest Italy) in the framework of the Italian National Environmental Radioactivity Network (RESORAD). The fallout samples (wet and dry deposition) are collected monthly by means of a stainless steel tank (surface area $\approx 4$ m$^2$) placed on the roof of the laboratory building (Figure 2) and always kept wet in order to avoid resuspension [32].

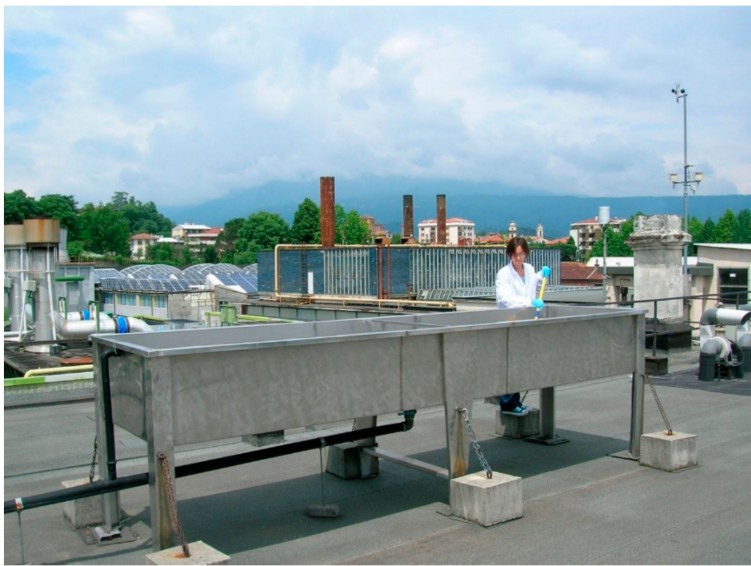

**Figure 2.** The stainless steel tank placed on the roof of the building for the collection of the wet and dry deposition samples.

Every month, at the end of the sampling period, the deposition is collected and dried. The residue is then weighed, placed in a small jar and measured by means of an hyperpure gamma-X (n-type) germanium detector (40% relative efficiency), able to detect the low energy 46.5 keV gamma emission of $^{210}$Pb (see Figure 3).

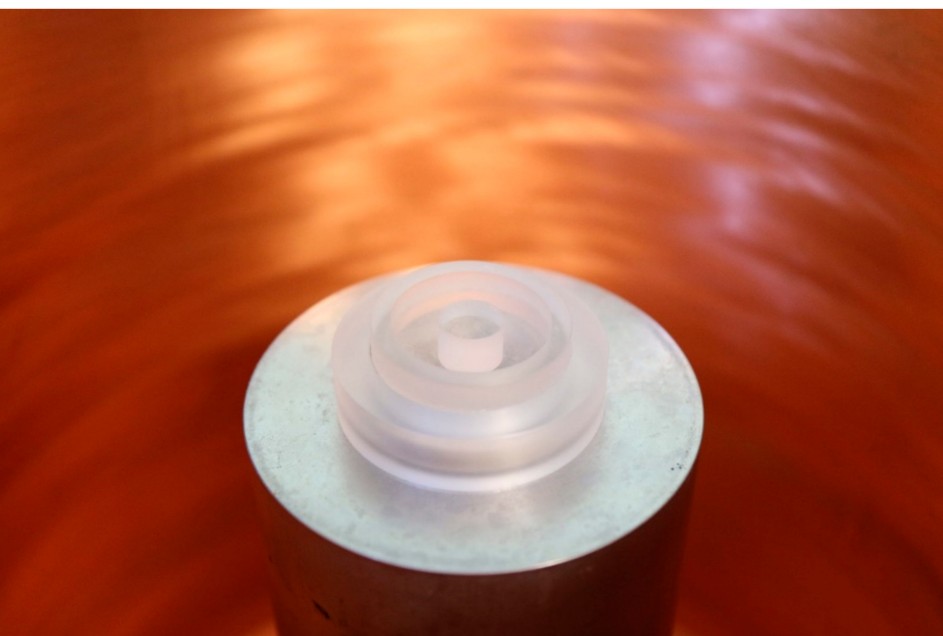

**Figure 3.** The little jar containing the 4 g of dry residue placed on to the top of HPGe n-type detector.

In order to have a standardized and calibrated counting geometry, a fixed quantity (4 g) of dry residue was put in the jar and uniformly distributed in a thin, cylindrically shaped geometry. As the photopeak efficiency was obtained by tracing with a multi-$\gamma$ standard calibration source 4 g of a soil-type material, no self-absorption corrections were needed.

In Figure 4, a typical $\gamma$ spectrum of a fallout sample is shown: marked in red, a well-shaped $^{210}$Pb $\gamma$ peak is clearly visible.

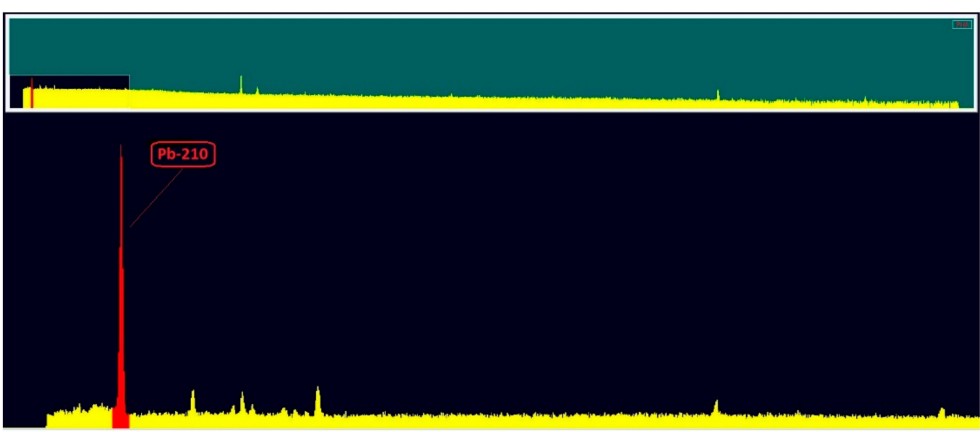

**Figure 4.** Typical $\gamma$ spectrum of a wet and dry deposition sample: the $^{210}$Pb 46.5 keV peak, marked in red, is well shaped and clearly visible.

Due to this long-lasting experimental work, a time series of the $^{210}$Pb wet and dry deposition in Ivrea, spanning the years 2005 to 2021, was available for this study.

The growth of $^{210}$Pb in the tank, due to the deposition processes, can be modelled by the following differential equation:

$$\frac{dA^{210Pb}}{dt} + \lambda_4 A^{210Pb} = \Phi_{210Pb} \tag{2}$$

In which $A^{210Pb}$ is the activity (Bq/m$^2$) accumulated in the tank, $F_{210Pb}$ is the $^{210}$Pb flux (wet and dry, expressed in term of Bq/m$^2$·s or Bq/m$^2$·h) while $\lambda_4$ is the $^{210}$Pb decay constant. The value of $F_{210Pb}$ depends of course on the quantity of $^{210}$Pb present in the atmosphere, i.e., the $^{210}$Pb atmospheric inventory, $A_4$. Being $\lambda_d$, the removal rate of $^{210}$Pb from the atmosphere, by definition the following relationship holds:

$$\Phi_{210Pb} = \lambda_d A_4 \tag{3}$$

Putting the right side of Equation (3) in Equation (2) and solving the differential equation, we obtain:

$$A^{210Pb} = \frac{\lambda_d A_4}{\lambda_4} \times \left(1 - e^{-\lambda_4 \times \tau}\right) \tag{4a}$$

where $\tau$ is the sampling time of the deposition measurements. Considering the typical value of the sampling time (1 month) and the numerical value of the $^{210}$Pb decay constant $\lambda_4$, we have $\lambda_4 \cdot \tau << 1$, and then the expression (4a) can be approximated to: $\times$

$$A^{210Pb} = \lambda_d \times A_4 \times \tau \tag{4b}$$

from which, taking into account for the last equation of system solution (1), the following equation for the radon flux, expressed in term of activity (Bq/m$^2$·s or Bq/m$^2$·h) may be written:

$$\Phi = \frac{(\lambda_1 + \lambda_d) \times (\lambda_2 + \lambda_d) \times (\lambda_3 + \lambda_d) \times (\lambda_4 + \lambda_d)}{\lambda_1 \lambda_2 \lambda_3 \lambda_4} \times \frac{A^{210Pb} \lambda_0}{\lambda_d \tau} \tag{5}$$

being $\Phi = \lambda_0 f$, where $\lambda_0$ is the radon decay constant.

However, for a robust evaluation of the radon flux, it is more convenient to take in Equation (5), instead of the highly variable $A^{210Pb}$ monthly values, the corresponding

asymptotic value $A_\infty^{210Pb} = \frac{\lambda_d D_4}{\lambda_4}$, obtained from (4a) as $\tau \to \infty$. Therefore Equation (5) becomes:

$$\Phi = \frac{(\lambda_1 + \lambda_d) \times (\lambda_2 + \lambda_d) \times (\lambda_3 + \lambda_d) \times (\lambda_4 + \lambda_d)}{\lambda_1 \lambda_2 \lambda_3 \lambda_4} \times \frac{A_\infty^{210Pb} \lambda_4 \times \lambda_0}{\lambda_d} \tag{6}$$

The $A_\infty^{210Pb}$ value in Equation (6) it is not a directly measurable quantity and needs to be evaluated. In principle, the estimation of the $A_\infty^{210Pb}$ value can be performed by means of a purely experimentally method, i.e., simply computing the accumulation of $^{210}$Pb on the Earth's surface by the series:

$$A_\infty^{210Pb} = \sum_{k=0}^{\infty} A_k^{210Pb} \times e^{-\lambda_4 \tau k} \tag{7}$$

in which the $A_k^{210Pb}$ is the experimentally measured monthly deposition values. The series should be of course truncated at a sufficiently large $k$, when the contribution of the last addendum becomes negligible. Unfortunately, because of the quite long half-life of $^{210}$Pb, the series converges quite slowly and therefore very long historical series of experimental data are likely necessary in order to obtain a reliable estimation of $A_\infty^{210Pb}$: for example, after 16 years (the timespan of our experimental data, 2005–2016), the value of the exponential factor of Equation (7) is still quite large: 0.61. To overcome this difficulty, a different approach can be followed. The individual monthly $A_k^{210Pb}$ values in Equation (7) are substituted with the average monthly value $\langle A^{210Pb} \rangle$, calculated in the whole time range considered in this study (2005–2021). Equation (7) can thus be rearranged to:

$$A_\infty^{210Pb} = \langle A^{210Pb} \rangle \times \sum_{k=0}^{\infty} e^{-\lambda_4 \tau k} \tag{8}$$

In this expression a geometric series appears, the sum of which can be easily calculated: $\sum_{k=0}^{\infty} e^{-\lambda_4 \tau k} = \frac{1}{(1 - e^{-\lambda_4 \tau})}$. Equation (6) of the radon flux can thus be rewritten in its final form:

$$\Phi = \frac{(\lambda_1 + \lambda_d) \times (\lambda_2 + \lambda_d) \times (\lambda_3 + \lambda_d) \times (\lambda_4 + \lambda_d)}{\lambda_1 \lambda_2 \lambda_3 \lambda_4} \times \frac{\langle A^{210Pb} \rangle \times \lambda_4 \times \lambda_0}{\lambda_d \times (1 - e^{-\lambda_4 \tau})} \tag{9}$$

## 3. Results

The experimental data considered in this study are gamma spectrometry measurements performed with HPGe detectors on the 186 wet and dry deposition samples collected monthly from October 2005 to April 2021. Of these, 46 had to be discarded because the gamma spectra were acquired using p-type 30% germanium detectors, resulting in $\gamma$ spectra with the $^{210}$Pb $\gamma$ emission peaks being very poor in shape and statistics. However, it is still possible to give a good estimation of the average deposition value with the remaining 140 available data, as the data are fairly uniformly distributed over the whole period (Figure 5).

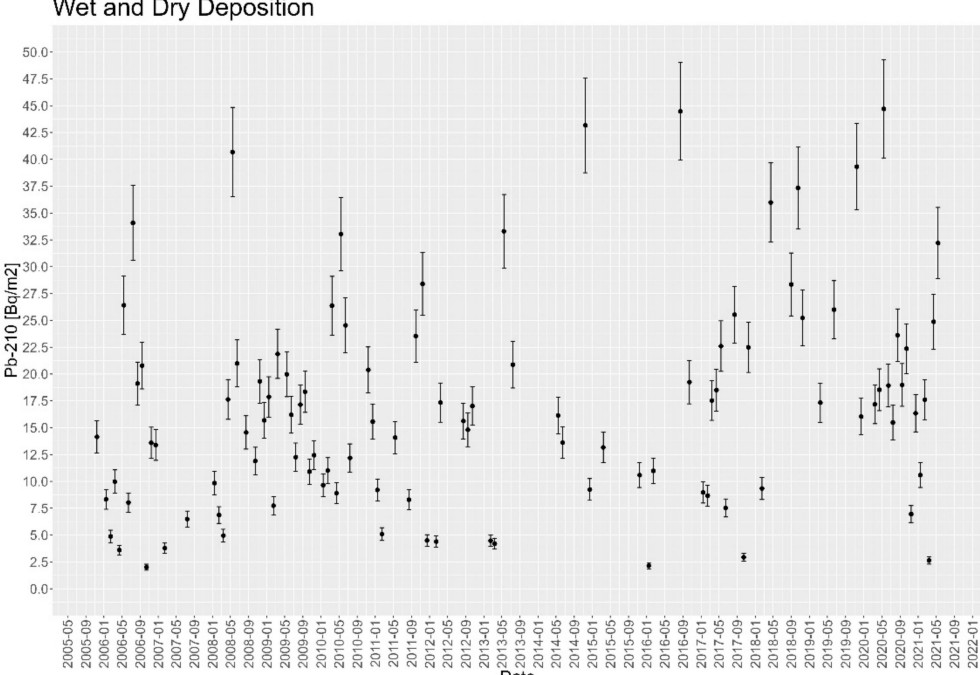

**Figure 5.** $^{210}$Pb wet and dry deposition data from 2005 to 2021, with their uncertainties ($2\sigma$).

The lowest measured value was 2.0 Bq/m$^2$ while the maximum was 44.7 Bq/m$^2$, the weighted mean value being $19.3 \pm 12.8$ Bq/m$^2$. The high observed variability and the resulting high value of the standard deviation may be easily explained by the very important role of the precipitation events: rain scavenges the atmosphere very effectively, thus substantially increasing the flux of $^{210}$Pb, being the airborne radionuclides attached to the sub-micron atmospheric particulate.

In order to investigate more deeply the relationship between the $^{210}$Pb flux and rain, the following simple deposition model can be used:

$$y = \alpha + \beta \times \left(1 - e^{-\gamma \times x}\right) \tag{10}$$

where $y$ represents the $^{210}$Pb deposition, the constant $\alpha$ at the right side of Equation (11) is the average contribution of the dry component while the second addendum is the wet component, depending on the amount of rain $x$ (mm) occurring during the sampling time.

The interpolation of the experimental data with the function given by Equation (10) is shown in Figure 6: it allows the estimation of the three free parameters of the model ($\alpha$, $\beta$, $\gamma$) and the related statistical parameters as well (see Table 1).

**Table 1.** Estimated coefficients and statistical parameters of the model.

| Parameters | Estimated Values |
|:---:|:---:|
| Model coefficient $\alpha$ | 7.80 Bq/m$^2$ |
| Model coefficient $\beta$ | 56.60 Bq/m$^2$ |
| Model coefficient $\gamma$ | 0.0021 mm$^{-1}$ |
| Pearson coefficient | 0.80 |
| Pearson coefficient CI (95%) | 0.71–0.86 |
| $^{210}$Pb estimated asymptotic value | 64.41 Bq/m$^2$ |

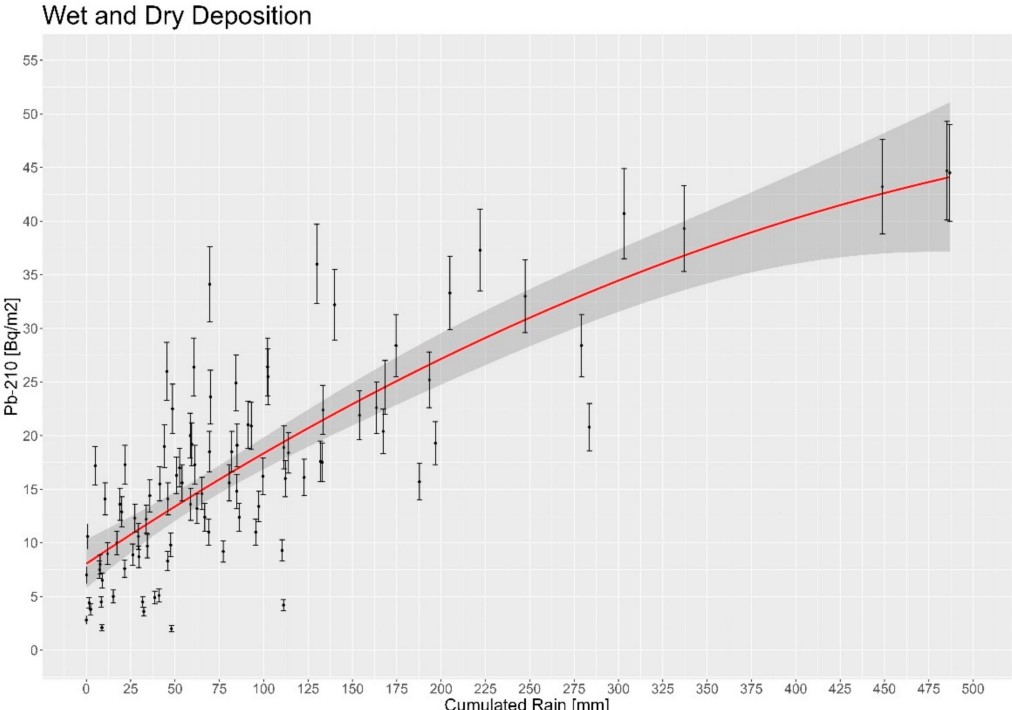

**Figure 6.** Interpolation (red line) of $^{210}$Pb total deposition as a function of the rain cumulated during the sampling time: the grey darker area indicates the interval of confidence of the curve (95%).

It is thus possible to evaluate, by means of Equation (10) the $^{210}$Pb average monthly deposition simply evaluating the expression at the corresponding average monthly precipitation value $\overline{x}$; remembering Equation (4b), the following holds:

$$\alpha + \beta \times \left(1 - e^{-\gamma \times \overline{x}}\right) = \lambda_d \times A_4 \times \tau \tag{11}$$

in which, two unknown quantities appear at the right side of the equation: the removal rate $\lambda_d$ and the $^{210}$Pb inventory $A_4$. However $A_4$ is nothing but the $^{210}$Pb estimated asymptotic value shown in Table 1. Therefore, from (11), the average $^{210}$Pb removal rate $\lambda_d$ can be calculated, giving: $\lambda_d = 3.69 \times 10^{-4}$ h$^{-1}$. Putting this number in flux Equation (9) together with all the other physical parameters we get the following value for the average radon flux: $\Phi = 57.8$ Bq/(m$^2\cdot$h). It can be observed that Equation (9) can be put in more simplified form: indeed, considering the numerical value just calculated for the removal rate $\lambda_d$ and the numerical values of the radionuclides decay constants, the following approximations hold:

$$(\lambda_1 + \lambda_d) \approx \lambda_1 \qquad (\lambda_2 + \lambda_d) \approx \lambda_2 \qquad (\lambda_3 + \lambda_d) \approx \lambda_3 \qquad (\lambda_4 + \lambda_d) \approx \lambda_d \qquad \left(1 - e^{-\lambda_4 \tau}\right) \approx \lambda_4 \tau$$

Equation (9) thus becomes:

$$\Phi = \frac{\langle A^{210Pb}\rangle \times \lambda_0}{\lambda_4 \times \tau} \tag{12}$$

in which appears only, as physical parameters, the $^{222}$Rn and $^{210}$Pb decay constants, $\lambda_0$ and $\lambda_4$, respectively. It is interesting to notice that average radon flux in the period 2005–2021 estimated using this latter approximate expression (12) gives a result almost identical to that obtained using the "exact" Equation (9): $\Phi = 57.1$ Bq/(m$^2\cdot$h).

All these estimations are of course affected by a quite large uncertainty whose evaluation is not a simple task. For instance, considering the standard deviation of the experimental $^{210}$Pb mean monthly deposition value, an apparently sound approach, would lead to a substantial and not realistic overestimation of the real uncertainty. In fact we would

have: $\langle A^{210Pb} \rangle = 19.3 \pm 12.8$ Bq/m$^2$, about 66%. Indeed, the very large standard deviation observed is mainly due to the different precipitation regimes affecting the monthly $^{210}$Pb deposition values rather than to a real variation in the radon flux $\Phi$ from the ground. Therefore, in order to reduce the variability due to precipitations, a new normalized variable is defined for any generic month *j*:

$$\langle A^{210Pb} \rangle_{Nj} = w_j \times \langle A^{210Pb} \rangle$$

where the normalization coefficients $w_j$ are given by:

$$w_j = \frac{D_j}{D(x_j)} \tag{13}$$

in which $D_j$ and $D(x_j)$ are, respectively, the experimental and the model estimated deposition data (Equation (11)), evaluated at the corresponding precipitation value $x_j$. The resulting average value of this new variable is nearly the same of the original one, while its standard deviation is appreciably reduced:

$$\langle A^{210Pb} \rangle_N = 21.6 \pm 10.8 \ \frac{\text{Bq}}{m^2}.$$

As a consequence of this, the final best estimation for the radon flux from $^{210}$Pb deposition data is $\Phi = 65 \pm 32$ Bq/(m$^2 \cdot$h). This result is in fairly good agreement with one of the few available radon flux measurements made in the Po Valley, Northern Italy (Facchini et al. [33]): $\Phi = 72$ Bq/(m$^2 \cdot$h). Other measurements, obtained by means of the exposure of passive nuclear track detectors in soil, were performed by our research group in the same area (Chiaberto E., Magnoni M., Righino F. [34]) and gave a somewhat greater average value, $\Phi = 97$ Bq/(m$^2 \cdot$h); however, these latter measurements were affected by a substantial overestimation, as no correction for the thoron interference was made at that time.

Regardless, the average radon flux value estimated in the present work matches quite well with globally accepted values. For example, Conen (2003 [35]), in a review article, gave a range between 15.1–75.6 Bq/(m$^2 \cdot$h) for the northern hemisphere, with an increasing trend from high (70° N) to low (30° N) latitudes; starting from this figure, a calculation for our latitude (45° N) leads to values very close to our result.

### 4. Conclusions

Monthly bulk deposition measurements of $^{210}$Pb performed by means of hyperpure germanium detectors in the framework of the Italian National Environmental Radioactivity Network (RESORAD) allowed the calculation of the average radon flux from soil. The value estimated with this method averaged over the time range considered (2005–2021) is $\Phi = 65 \pm 32$ Bq/(m$^2 \cdot$h)—in quite good agreement with published data, obtained using very different techniques. The proposed method is thus proved to be reliable and useful for radon flux estimation. As $^{210}$Pb deposition measurements are routinely performed all over the world in many environmental radioactivity monitoring networks, this method could be used elsewhere, helping researchers to obtain a worldwide map of the average radon flux, provided that a sufficiently long and reliable historical time series (at least 15–20 years) of $^{210}$Pb data is available.

**Author Contributions:** Conceptualization, M.M.; methodology, M.M., L.B.; software, L.B., S.B.; experimental work, L.B., B.B., S.B.; data curation, M.M., L.B., S.B.; writing—original draft preparation, M.M.; writing—review and editing, M.M., E.C. All authors have read and agreed to the published version of the manuscript.

**Funding:** This research received no external funding.

**Institutional Review Board Statement:** Not applicable.

**Informed Consent Statement:** Not applicable.

**Data Availability Statement:** The experimental data reported in this study can be partially found in the official report of ARPA Piemonte, published in the website: www.arpa.piemonte.it, accessed on 31 March 2022.

**Conflicts of Interest:** The authors declare no conflict of interest.

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
