# Peer review of "Evaluation of the Terrestrial 222Rn Flux from 210Pb Deposition Measurements"

_environments, doi:10.3390/environments9060068_

Round 1

Reviewer 1 Report

The Ms, titled "Evaluation of the terrestrial 222Rn flux from 210Pb deposition measurements," presents a new method for indirectly calculating the average value of the terrestrial radon flux, with the authors deriving this parameter from a historical series of 210Pb spectrometry deposition measurements conducted within the RESORAD network. The text is in excellent condition and sounds sintiflicy. After a few comments, the MS may be published in an environments journal.

The most important findings should be included in the abstract, and the conclusion should back up the main hypotheses with recommendations.

Figure 4 requires a thorough examination, as well as a higher resolution if necessary.

Equations should be formatted in the same way.

Author Response

Thank you for your observation and remarks. In the following a point by point answer, marked in red

The Ms, titled "Evaluation of the terrestrial 222Rn flux from 210Pb deposition measurements," presents a new method for indirectly calculating the average value of the terrestrial radon flux, with the authors deriving this parameter from a historical series of 210Pb spectrometry deposition measurements conducted within the RESORAD network. The text is in excellent condition and sounds sintiflicy. After a few comments, the MS may be published in an environments journal.

The most important findings should be included in the abstract, and the conclusion should back up the main hypotheses with recommendations.

The most relevant findings are now included in the abstract

Figure 4 requires a thorough examination, as well as a higher resolution if necessary.

Errors bar added to the experimental data

Equations should be formatted in the same way.

Equations formatting armonized

Reviewer 2 Report

The methodological approach of the described research is sound; the measurement constraints related to low energy (46.5 kV) gamma measurements are critically discussed, the statistical treatment of the data series is convincing.

My only point of criticism is the somehow spare discussion of previous work dealing with excess lead deposition; e.g.  like the attached files. I would recommend to revise the introduction in this regard.

Author Response

Thank you for your comments and suggestions

The methodological approach of the described research is sound; the measurement constraints related to low energy (46.5 kV) gamma measurements are critically discussed, the statistical treatment of the data series is convincing.

My only point of criticism is the somehow spare discussion of previous work dealing with excess lead deposition; e.g.  like the attached files. I would recommend to revise the introduction in this regard.

Excess lead studies mentioned and cited in the revised introduction

Reviewer 3 Report

I have read with interest the paper, which present the try to estimate radon flux by measuring low energy gamma-ray emitted in the subsequent nucleus in the radon decay-chain. Topic is interesting, and wort to be publish, however, I would like to ask some questions – mainby (b) and (c),  that may improve presentation of the results.

a)
Line 61 of the manuscript state half-life of 210Pb to be 22.23 years, while in the Figure 1 it is 22,3 a.
Please check the values in the Figure 1 with the well evaluated data from NNDC (vide https://www.nndc.bnl.gov/nudat3/getdatasetClassic.jsp?nucleus=210Pb is giving half-life of 210Pb to be equal 22.20(22) years).
Please change the abbreviations in the Figure 1 from Italian to English (a->y; g-> h) and comma to decimal dots.

b) Just to be precise: 46.5 keV gamma-ray is emitted by de-excitation of 0- excited state of 210Bi (that excited state has half-life less than 3 ns) aftter beta decay of 210Pb. 
The same is said in the line 124:
the low energy 46.5 keV gamma emission of 210Pb. 
While to be precise it is: 46.5 keV gamma emission of excited state of 210Bi feed by beta decay of 210Pb.

c) Main question(s):
In the chapter 2, method to determine Radon flux by measuring 210Pb activity is well described.
However, in the chapter 3 of the Results, authors do not wrote description of the methods of determination of the 210Pb activity - how this procedure was performed? How it was corrected by self-absorption of gamma-rays?

This is key point to determine the flux of radon, then please even provide one figure with the sample quality of gamma-ray registered energy spectrum with 46.5 keV gamma-ray line.

Are there assigned errors for every point in the Figure 3? As, for the 186 samples 46 were to be discarded, then for the others errors of the activity should be quite visible, an usable in the mean value estimation. 
What was the factor to eliminate 46 samples from the dataset - no visible gamma-ray peak, bad quality, or other?

Does it change the mean and its error to take into account every point error bars (to be used for weighting that data points)? (line 200)

How it will look like Figure 4 with error bars added for every Pb-210 activity point? Does it change fitted parameter values from Table 1?Please check, if abovementioned remarks will change the extracted radon flux value (line 260). 

Author Response

Thank you very much for your comments and remarks: they helped us a lot to improve the paper.

In the following a point by point answer

I have read with interest the paper, which present the try to estimate radon flux by measuring low energy gamma-ray emitted in the subsequent nucleus in the radon decay-chain. Topic is interesting, and wort to be publish, however, I would like to ask some questions – mainby (b) and (c),  that may improve presentation of the results.

a)
Line 61 of the manuscript state half-life of 210Pb to be 22.23 years, while in the Figure 1 it is 22,3 a.
Please check the values in the Figure 1 with the well evaluated data from NNDC (vide https://www.nndc.bnl.gov/nudat3/getdatasetClassic.jsp?nucleus=210Pb is giving half-life of 210Pb to be equal 22.20(22) years).
Please change the abbreviations in the Figure 1 from Italian to English (a->y; g-> h) and comma to decimal dots.

Thanks for the remarks. Changes were made in Figure 1 accordingly with the text. Italian half lives abbreviations substituted with English ones.

  1. b) Just to be precise: 46.5 keV gamma-ray is emitted by de-excitation of 0- excited state of 210Bi (that excited state has half-life less than 3 ns) aftter beta decay of 210Pb. 
    The same is said in the line 124:
    the low energy 46.5 keV gamma emission of 210Pb. 
    While to be precise it is: 46.5 keV gamma emission of excited state of 210Bi feed by beta decay of 210Pb.

You’re right of course, thanks. Text corrected accordingly with the remarks.

  1. c) Main question(s):
    In the chapter 2, method to determine Radon flux by measuring 210Pb activity is well described.
    However, in the chapter 3 of the Results, authors do not wrote description of the methods of determination of the 210Pb activity - how this procedure was performed? How it was corrected by self-absorption of gamma-rays?

In order to have  a standardized and calibrated counting geometric a fixed quantity (4 g) of dry residue is put in the jar and uniformly distributed in a thin cylindrical shaped geometry. As the photopeak efficiency was obtained by tracing with a multi-g standard calibration source a soil-type material, no self absorption correction are needed. (Figure and explanation added to the text, from line 132 to line 140)

This is key point to determine the flux of radon, then please even provide one figure with the sample quality of gamma-ray registered energy spectrum with 46.5 keV gamma-ray line.

Figure of the gamma-ray 46.5 gamma peak added

Are there assigned errors for every point in the Figure 3? As, for the 186 samples 46 were to be discarded, then for the others errors of the activity should be quite visible, an usable in the mean value estimation. 
What was the factor to eliminate 46 samples from the dataset - no visible gamma-ray peak, bad quality, or other?

The reason of rejecting the data of 46 samples out of 186 was either the bad quality of  the gamma-ray peak and/or the quite large uncertainty of the photopeak efficiency due to the fact that for these 46 samples the gamma spectra were obtained using a p-type 30% HPGe detector. For this type of detector, due to the very large steepness of the efficiency curve, the photopeak efficiency at 46.5 keV was extrapolated from the lowest energy gamma peak of the reference calibration source, the 59.5 keV peak from Am-241. Please note that the fallout measurements were originally not specifically focused on the 210Pb detection; they were originally performed in the framework of a routinely monitoring program and therefore, in some cases, the more easily available detector type was used.

Does it change the mean and its error to take into account every point error bars (to be used for weighting that data points)? (line 200)

Taking into account every error bars the mean changes slightly, from 18.4 to 19.3 (see text)

How it will look like Figure 4 with error bars added for every Pb-210 activity point? Does it change fitted parameter values from Table 1?Please check, if abovementioned remarks will change the extracted radon flux value (line 260).

The error bar to the deposition data were added in Figure 3 and 4. We have performed the calculations taking into account for the error bars (weighted mean and data interpolation, Table 1). Considering these new data the 210Pb averages are slightly changed and the final radon flux estimation as well: from 57 to 65 Bq/m2*h